# MIM-CyCIF: masked imaging modeling for enhancing cyclic immunofluorescence (CyCIF) with panel reduction and imputation

Zachary Sims[1], Gordon B. Mills [2] & Young Hwan Chang [1,2] ✉

Cyclic Immunofluorescence (CyCIF) can quantify multiple biomarkers, but panel capacity is limited by technical challenges. We propose a computational panel reduction approach that can impute the information content from 25 markers using only 9 markers, learning co-expression and morphological patterns while concurrently increasing speed and panel content and decreasing cost. We demonstrate strong correlations in predictions and generalizability across breast and colorectal cancer, illustrating applicability of our approach to diverse tissue types.

Emerging Multiplexed Tissue Imaging (MTI) platforms[1–5] produce rich, spatially resolved protein expression information that enables analysis of tissue samples at single cell resolution[6–12]. However, the broad application of existing MTI platforms in cancer research and clinical diagnosis is hindered by high material costs, data storage requirements, and the need for specialized equipment and technical expertise to mitigate experimental variabilities. Moreover, the number of markers within MTI panels is limited by cost and time constraints encompassing image acquisition, marker selection, and validation. Tissue degradation through repeated staining and removal cycles adds to this challenge[4]. Thus, the selection of markers for the panel becomes crucial to interrogate a wide spectrum of cell states and phenotypes[5,13,14].

Previous studies computationally optimized MTI panel reduction and marker prediction. Ternes et al.[15] pioneered Cyclic Immunofluorescence (CyCIF) panel reduction and imputation using a two-step approach: exploring multiple strategies for marker selection and training a multi-encoder variational autoencoder (ME-VAE)[11] to reconstruct the full 25-plex CyCIF images at the single cell level. Wu et al. proposed a three-step method using a concrete autoencoder and convolution neural network to reduce CODEX markers and predict intensity via a regression model[16]. Sun et al. iteratively trained a U-Net to reconstruct patch-level images, aiding marker selection for a reduced panel[17]. In contrast to prior research, where panel selection is separate from full panel reconstruction, our method integrates iterative marker selection within a pre-trained model, streamlining panel reduction and reconstruction. Unlike fixed-size reduced panels, our method optimizes marker selection during inference, enhancing efficiency, reliability, and practicality for model training.

Inspired by the success of masked language modeling in natural language processing[18], the concept of masked image modeling (MIM) has gained traction in computer vision[19,20]. MIM-based models resemble denoising autoencoders[21], utilized for data restoration and model pretraining. Nevertheless, the utilization of masked image modeling for missing data imputation tasks has been minimally investigated.

## Results

Employing a self-supervised trained masked autoencoder (MAE), we reconstruct masked CyCIF image channels at the single-cell level and identify optimal reduced panel sets for complete panel reconstruction. Through the architecture and masked token prediction task outlined in Fig. 1A, we demonstrate successful imputation of CyCIF image channels at the single-cell level through 'channel in-painting'. Our model takes in a collection of single-cell images, each containing 25 channels representing individual CyCIF marker stains. During training, we set a fixed ratio of channels that will undergo random masking for each sample (Fig. 1B left). Our model is then tasked with reconstructing these masked channels (Fig. 1B right). After model training, the selection of masked channels becomes feasible to determine the optimal unmasked channel combination for accurate reconstruction of masked counterparts. This strategy is harnessed to progressively curate an enhanced marker panel (Fig. 1C).

In each iteration, the marker selection is the one maximizing the Spearman correlation between actual and predicted mean intensities for the remaining held-out markers (Methods: Iterative Panel Selection). Subsequently, this iterative process of refining panels establishes an order of

[1]Department of Biomedical Engineering and Computational Biology Program, Oregon Health & Science University, Portland, OR, USA. [2]Knight Cancer Institute, Oregon Health & Science University, Portland, OR, USA. ✉e-mail: chanyo@ohsu.edu

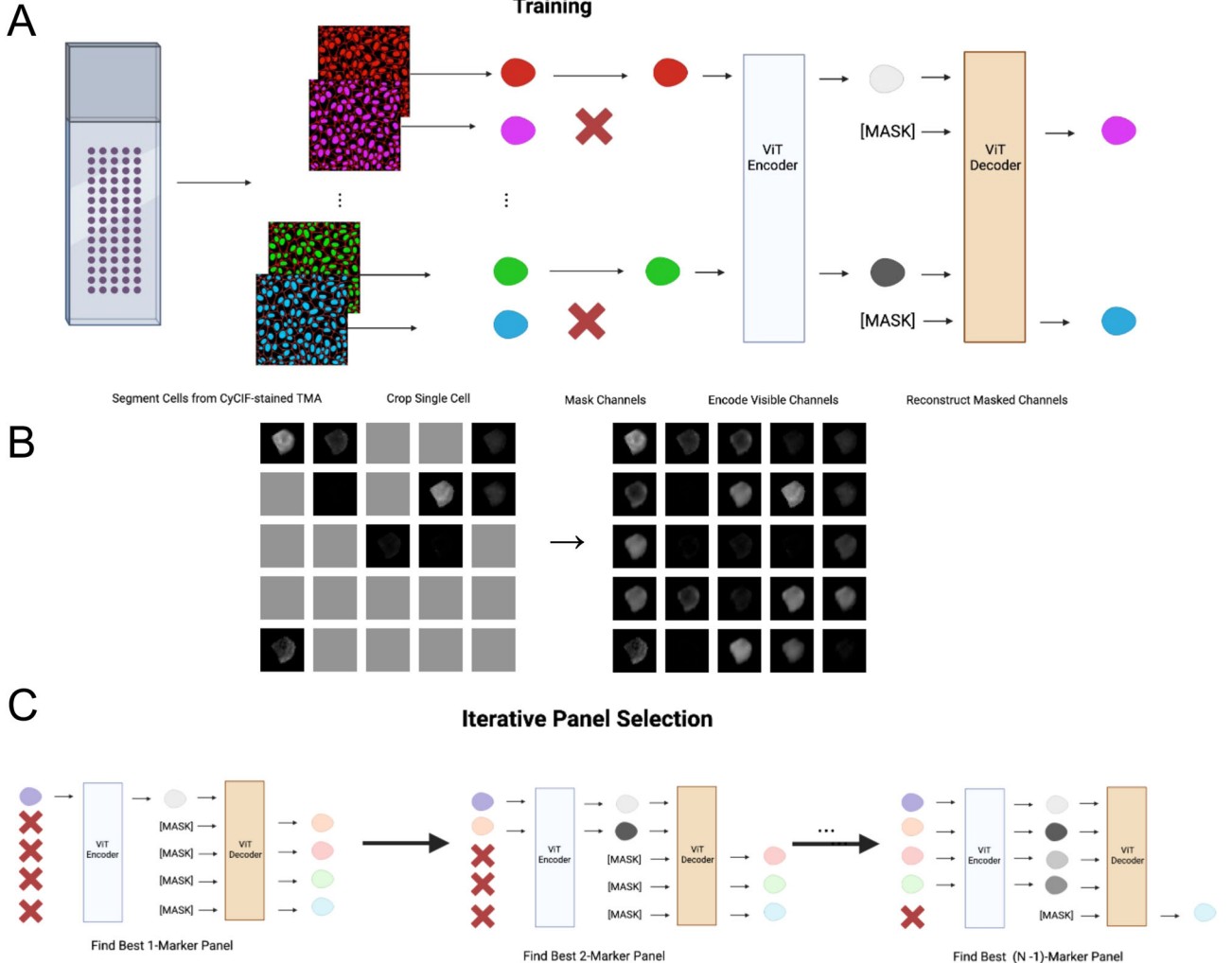

**Fig. 1 | Masked autoencoder for panel reduction and marker imputation.**
**A** Model architecture: CyCIF image-derived single cells undergo channel-wise masking followed by the encoding of unmasked channels using a Vision Transformer (ViT). A distinct mask token represents masked channels. A ViT decoder then reconstructs the masked channels, completing the image reconstruction process. **B** CyCIF channel-wise masking (left) and reconstruction (right): 25-channel images arranged into a 5 × 5 grid format, facilitating conversion from a patch-wise masking strategy into a channel-wise masking strategy. **C** Iterative marker selection: leveraging the trained model, an optimal marker order is established by gradually increasing the panel size. Each step selects the next marker based on its ability to maximize the Spearman correlation between actual and predicted mean intensity for masked channels. This refines marker panel ordering, enhancing prediction accuracy. Parts of Fig. 1A were created using BioRender (www.biorender.com).

markers that roughly represents predictive value regarding other markers (Fig. 2A). Visualized in columns, the chosen markers' influence on predicting withheld markers is depicted, while rows illustrate the corresponding improvements in prediction for each withheld marker.

In our prior study[15], we pre-selected optimally reduced panels comprising 3, 6, 9, 12, 15, and 18 markers through Spearman Correlation. The subset of markers exhibiting the highest correlation with the remaining withheld markers was chosen as the reduced panel. Figure 2B illustrates the enhancement in prediction resulting from the replacement of ME-VAE with MAE. With the same 9-marker reduced panel, MAE improves the average correlation of withheld marker predictions by 0.22 (yellow violin plot). A further enhancement is achieved by utilizing the reduced panel generated via iterative selection, yielding an additional 0.01 improvement (green violin plot). Figure 2C describes the comparison between real and predicted single-cell mean intensity values across different panel sizes (3, 6, 9, 12, and 15 markers) using plots. As the reduced panel size increases, we observe an increase in prediction correlation. We also evaluate the structural similarity index measure (SSIM) (Methods: Model's Performance Evaluation), a widely adopted metric capturing image similarity as perceived by the human visual system[22], between real and predicted single-cell expression at the pixel level (Fig. 3A). Example reconstructions of image channels are depicted in Fig. 3B, C.

In addition, we demonstrate similar effectiveness on the colorectal cancer (CRC) tissue microarray (TMA) dataset (Supplementary Figs. 1 and 2). Moreover, the model generalizes well to unseen data within the same batch, demonstrated by conducting 5-fold cross-validation across the CRC TMA cores. Cross-validation was performed on the CRC TMA dataset to evaluate model performance on unseen data. We divide the dataset at the TMA core level by separating the cores into 5 sets of 64 TMA cores for the test sets and the remaining TMA cores are used to train 5 separate models. As TMAs typically encompass multiple patients, this test effectively demonstrates the model's generalizability. The performance of the 5 models on different reduced panel sizes is shown in Supplementary Fig. 3. By splitting the data at the core level, we show that our model generalizes well across different patient samples.

To further demonstrate model generalizability to samples stained in a different batch, we test the CRC model on a whole-slide image (WSI) stained using the same panel. Because three of the cores in the TMA were derived from the tissue section in the WSI, we can directly assess the model's robustness to batch effect while preserving intra-patient intensity

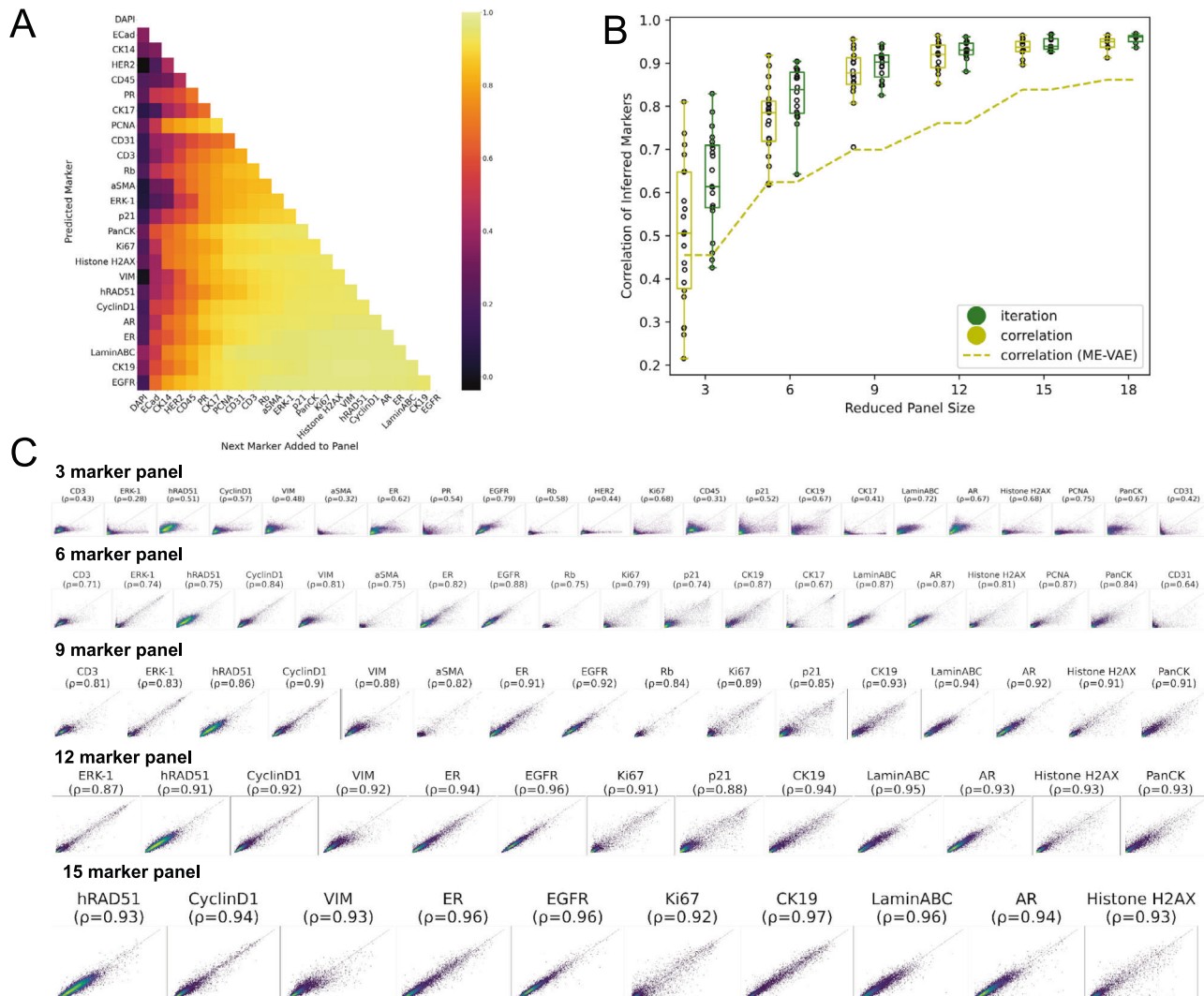

**Fig. 2 | Model Evaluation. A** Impact of individual markers: Depicting the effect of marker selection on the prediction of specific marker intensities. Each row tracks the improvement in prediction for a specific marker as new markers are added to the reduced panel. A heatmap illustrates the mean Spearman correlation between actual and predicted mean intensities. **B** Comparison to prior work[15]: The yellow dashed line shows the mean Spearman correlation achieved for predicted marker intensities utilizing ME-VAE. The corresponding yellow violin plot demonstrates the performance of MAE on the same reduced panels that showed the optimal results in Ternes et al.[15]. Green violin plot showcases MAE performance using reduced panels selected using the iterative panel selection approach. **C** Real versus predicted single-cell mean intensity values. Plots of actual versus predicted single-cell mean intensity values are presented for reduced panel sizes of 3,6,9,12, and 15 markers, respectively. A random subset of 10,000 cells is shown. The Spearman correlation for each marker is indicated.

distributions. Supplementary Fig. 4 shows the performance of the model, trained on the CRC TMA with reduced panels selected from the TMA, on this dataset. Although the model exhibits a modest performance reduction (0.26 reduction in Spearman correlation using the same 9 marker panel), these results hold promise, particularly given potential limitations in generalizability stemming from ROI selection bias and a single TMA reference for batch correction. A histopathologist would heavily favor tumor regions for core punchouts, whereas the full WSI contains a more heterogeneous tissue region. Therefore, markers that are not expressed in tumor cells are potentially underrepresented in our training set. This finding aligns with prior findings on small TMA cores, emphasizing the enhanced representation provided by randomly selected multiple TMA cores in capturing tumor or immune contexture as compared to WSI[23–25].

## Discussion

Our study highlights the efficacy of utilizing MAE to generate high-plex CyCIF data from only a few experimental measurements, significantly reducing the required biomarkers to interrogate a sample. The ability to identify a biomarker subset and perform in silico prediction offers several advantages. Our method empowers users to access a more extensive set of biomarkers beyond those experimentally measured. Additionally, it enables the allocation of resources for the exploration of novel biomarkers, thereby enhancing cell type differentiation and disease characterization. Furthermore, it can manage instances of assay failures such as low-quality markers, technical noise, and/or potential tissue loss in later CyCIF rounds. It also can artificially up-sample and incorporate additional panel markers.

In this current study, we present advancements in upstream performance that are anticipated to yield downstream benefits, specifically enhancing population-scale assessment and spatial interaction. We avoid directly comparing and classifying cell types in subsequent analysis to avoid oversimplifying complex cellular phenotypes. This improvement stems from a reduction in modeling errors at the single-cell level within our proposed approach. It is important to note that cell type determination in MTI settings involves diverse methodologies and can be influenced by factors like imperfect cell segmentation, marker selection, data preprocessing and normalization[26–28], and algorithm choice[29]. It is also noteworthy that a recent investigation by Wu et al.[16] achieved promising outcomes, exhibiting an average Pearson correlation coefficient (PCC) of 0.534 for all

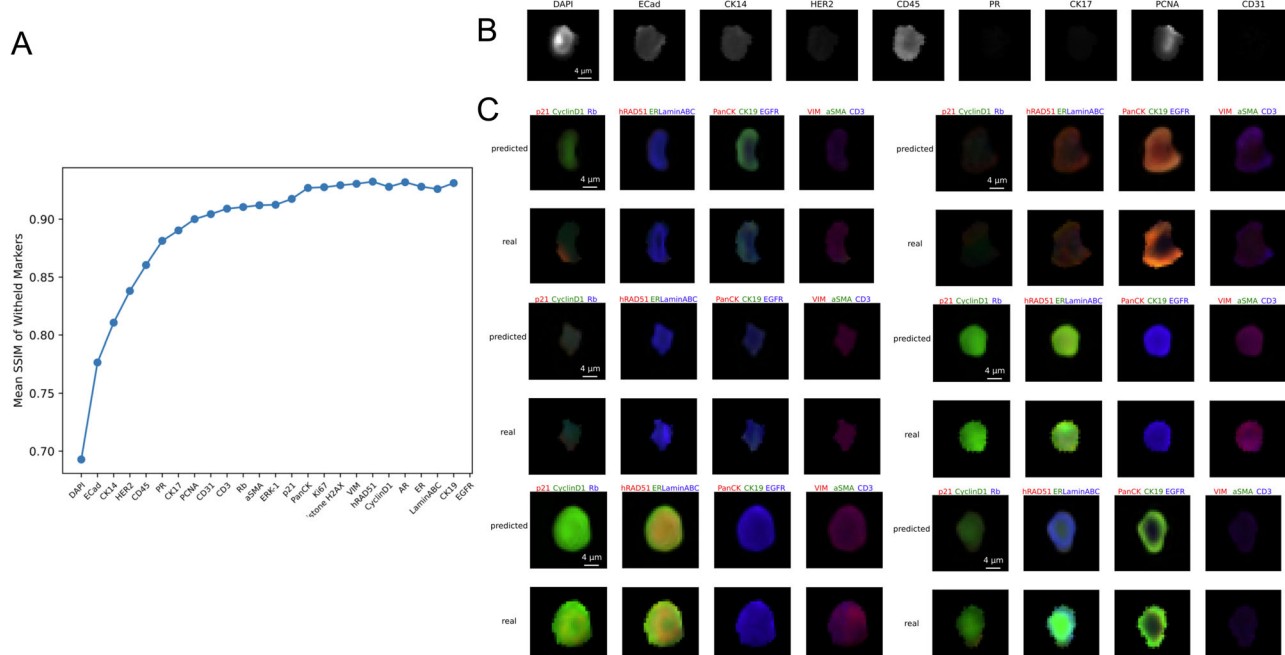

**Fig. 3 | Structural Similarity Index and Pixel-level Reconstructions. A** The structural similarity index for the BC TMA. **B** Example reconstruction of a single-cell image using a 9-marker reduced panel. Top row shows the image channels for the markers in the reduced panel. Bottom two rows show real and predicted pixel values for the imputed marker channels. **C** Examples show multiple predicted channels overlaid.

predicted biomarkers. Their study emphasized the significance of imputed biomarkers in accurately classifying cell types, as evidenced by an F1 score of 0.727, and predicting patient survival outcomes. Our objective is to further elevate our model's performance, aiming for a correlation score of 0.9, surpassing existing standards. This aspiration is grounded in our results, which have already demonstrated performance that exceeds accepted benchmarks for biomarker prediction accuracy.

While our work demonstrates a promising framework for unifying panel reduction and marker imputation into a single model, we emphasize that our results serve as a proof-of-concept study and acknowledge the limitations that may prevent our method from being incorporated into traditional biological studies, such as error propagation from single-cell intensities down to population-scale assessments. Another limitation is that we only evaluate our model on one MTI platform (CyCIF). Assessing the generalizability of our trained model to other platforms is an important step toward leveraging our model to democratize MTI. In future work, we will address these limitations by exploring different normalization strategies[26–28] to reduce marker intensity variability across batches. Additionally, we will explore a more diverse training dataset, incorporating WSIs in different batches, effectively mitigating TMA sampling bias and batch effects. This expanded dataset will also allow for a rigorous analysis of downstream performance.

## Methods
### CyCIF image dataset
Our data consists of CyCIF images from two TMA datasets, one containing breast cancer (BC) tissue and the other containing colorectal cancer (CRC) tissue. The tissue microarrays (TMAs) CyCIF imaging data are available via HTAN (https://humantumoratlas.org/). The breast cancer (BC) TMA contains 88 cores representing 6 cancer subtypes. The colorectal cancer (CRC) TMA contains 332 cores. Biomarker panels for the two TMAs are shown in Supplementary Table 1.

### Image preprocessing
The original CyCIF marker intensities (16 bit) were rescaled to the 8 bit image ([0,255] range). For the BC TMA, we simply used preprocessed data

in our previous work[15]. For the CRC TMA, any core containing a channel with a mean intensity beyond 2 standard deviations from the mean intensity of the entire TMA for that channel was dropped. This resulted in 12 cores being removed. For the CRC WSI, to mitigate batch-to-batch staining variations, the intensity distributions for each channel in the WSI were normalized using histogram matching with 3 cores from the CRC TMA - obtained from the same tissue section as the WSI. Individual cores and the WSI were then segmented using MESMER[30]. We use the whole cell masks generated using the max projection of the PanCK and CD45 channels as the membrane marker to crop each cell down to a $32 \times 32$ pixel region. The background of each single-cell image is then zeroed out, and the polar axis and center of mass are aligned. This resulted in 742,169 cells for the CRC TMA, 742,799 cells for the CRC WSI, and 691,893 cells for the BC TMA. We use 90% of the cells randomly selected as the training set and withhold 5% for validation/panel selection and 5% for testing.

### Masked image modeling
We modify the patch-wise masking strategy in MAE[19] to a channel-wise masking strategy. This involves resizing the $32 \times 32 \times 25$ multichannel single-cell image ($32 \times 32$) to a $5 \times 5$ grid format (Fig. 1B), creating a resulting image size of $160 \times 160$ (i.e., $(32 \times 5) \times (32 \times 5)$) with a single channel. Consequently, the patch size for MAE is adjusted to $32 \times 32$, where each patch now corresponds to an individual channel within the original multi-channel image.

We use a Vision Transformer (ViT) encoder and a ViT decoder as in MAE, both set to 8 heads and 6 layers, and a 2048-dimension multilayer perceptron layer. The embedding dimensions were 1024 for the encoder and 512 for the decoder. We train the model using 8 Nvidia A40 GPUs for 300 epochs using a batch size of 4096, Adam optimizer and a learning rate of 1e-3.

Although the trained model works on a range of reduced panel sizes, during training the number of channels to be masked is set to a fixed ratio. We evaluate different masking ratios for training by assessing the performance of different reduced panel sizes in inference on the BC TMA dataset. For testing, we choose the optimal reduced panels identified in Ternes et al.[15], which have sizes of 3,6,9,12,15, and 18 markers (88%, 76%, 64%, 52%,

40%, and 28% masking ratios, respectively). We train three models using a fixed masking ratio of 25%, 50%, and 75%, and find that the 50% masking ratio results in the best overall performance across different panel sizes in inference (Supplementary Fig. 5).

## Iterative panel selection

To obtain optimal reduced panel sets, we leverage the trained model to determine which markers are most informative. To do this, we iteratively determine an ordering of markers such that the first $k$ markers result in the best reconstruction of the remaining $n - k$ markers, measured by the Spearman correlation of the predicted mean intensity at the single cell level. We start with $k = 2$, setting the first marker to be DAPI, as nuclear staining is important for downstream analysis such as registration as well as determining cell morphology. We then iterate through the remaining 24 markers to determine which marker, along with DAPI, results in the best reconstruction of the remaining 24 markers (Figs. 1C and 2A). We repeat this process until we find the best $k = 24$ marker panel:

$$Panel_1 = \{c_{DAPI}\}$$

$$Panel_k = Panel_{k-1} \cup \left\{ \arg\max_c \left( \rho_{Y, f(X, Panel_{k-1} \cup \{c\}, \theta)} \right) \right\}$$

Where $c$ is the marker channel being considered for inclusion into $Panel_k$, $Y$ is the set of ground truth masked channels, $X$ is the set of unmasked channels, $f$ is the trained MAE model parameterized by $\theta$, which returns the reconstructed masked channels, and $\rho$ is the Spearman correlation between the mean intensities of $Y$ and the output of $f$.

## Model's performance evaluation

We evaluate the model's performance using Spearman correlation. We assess the agreement between actual and the predicted mean marker intensities within the cell boundaries. This analysis is crucial as it quantifies specific cellular component expression levels across markers, characterizing cellular phenotypes. To address potential concerns about the sensitivity of Spearman correlation to differences in predicted intensities, we computed the correlation variance across stains. This approach allows us to assess the consistency of stain predictions, ensuring that the model performs well across all stains rather than excelling in some while performing poorly in others. By considering correlation variance, we aim to provide a more nuanced understanding of the model's predictive capabilities.

In addition to Spearman correlation, we incorporated the Structural Similarity Index Measure (SSIM) to assess the quality of reconstructed CyCIF images. SSIM is a standard metric for tasks such as image translation, denoising, and restoration, as it evaluates both intensity and spatial information. This dual evaluation approach enhances the comprehensiveness of our assessment, addressing concerns about potential noise, background interference, or extreme outliers influencing the results.

## Statistics and reproducibility

We calculate the mean of the Structural Similarity Index (SSIM) and Pearson correlation coefficients for each marker. This approach ensures a robust statistical evaluation of the dataset, providing insights into the consistency and reliability of the observed trends across different markers. We also provide all the code necessary to ensure reproducibility of our results.

## Reporting summary

Further information on research design is available in the Nature Portfolio Reporting Summary linked to this article.

## Data availability

As part of this paper all images at full resolution, all derived image data (e.g. segmentation masks), and all cell count tables will be publicly released via the NCI-recognized repository for Human Tumor Atlas Network (HTAN; https://humantumoratlas.org/) at Sage Synapse (associated Identifiers: HTAN TNP – TMA, OHSU_TMA1_004-XX) where XX represents TMA core ID. For CRC TMA, see source publication[24] (https://doi.org/10.1016/j.cell.2022.12.028). All the source data for graphs and figures are available at the following link[31]: https://doi.org/10.5281/zenodo.10724928.

## Code availability

All software used in this manuscript is detailed in the article's Methods section and its Supplementary Information. The associated scripts[32,33] are freely available via GitHub as described at https://github.com/zacsims/IF_panel_reduction (https://doi.org/10.5281/zenodo.10835282).

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

## Acknowledgements
We thank Jerry Lin, Yu-An Chen, and Peter K. Sorger (Harvard Medical School) for sharing data and providing useful feedback. This work was carried out with major support from the National Cancer Institute (NCI) Human Tumor Atlas Network (HTAN) Research Centers at OHSU (U2CCA233280). Y.H.C. is supported by R01 CA253860 and Kuni Foundation Imagination Grants. The resources of the Exacloud high-performance computing environment developed jointly by OHSU and Intel and the technical support of the OHSU Advanced Computing Center are gratefully acknowledged.

## Author contributions
Conceptual development: Z.S., G.B.M, Y.H.C. Data analysis and computational implementation: Z.S. Manuscript writing: Z.S., Y.H.C.

## Competing interests
The authors declare the following competing interests: G.B.M. is a SAB member or Consultant: for Amphista, Astex, AstraZeneca, BlueDot, Chrysallis Biotechnology, Ellipses Pharma, GSK, ImmunoMET, Infinity, Ionis, Leapfrog Bio, Lilly, Medacorp, Nanostring, Nuvectis, PDX Pharmaceuticals, Qureator, Roche, Signalchem Lifesciences, Tarveda, Turbine, Zentalis Pharmaceuticals. G.B.M. has Stock/Options/Financial relationships with: Bluedot, Catena Pharmaceuticals, ImmunoMet, Nuvectis, SignalChem, Tarveda, and Turbine. G.B.M. has Licensed Technology: HRD assay to Myriad Genetics, DSP patents with Nanostring. G.B.M. has Sponsored research with AstraZeneca. The other authors declare no competing interests.
