## [Peer Review File · Communications Biology]

Reviewers' comments:

Reviewer #1 (Remarks to the Author):

CyCIF and other multiplexed techniques offer the potential to measure numerous biomarkers, yet its panel capacity faces constraints due to technical obstacles. To address this limitation, the authors here introduced a computational method for panel reduction. This approach can effectively deduce information from markers employing just 9 key markers. It accomplishes this by learning co-expression and morphological patterns, resulting in enhanced speed, expanded panel content, and reduced costs. Though the idea and implementation here are both legit and solid, there are serious issues here need to be addressed.

First, the evaluation of model performance was conducted solely at the single-cell level, lacking a population-scale assessment. Furthermore, since the primary purpose of CyCIF and other multiplex imaging approaches is to investigate spatial interactions and structures within tissues, modeling errors at the single-cell level could potentially magnify in these aspects. The authors should endeavor to address these caveats, such as comparing the overall distribution of markers between predicted and actual values.

Secondly, the use of Spearman correlation here may not be entirely appropriate. Predicted intensities could differ significantly from the ground truth while still achieving a high Spearman correlation. Moreover, this correlation might be influenced by noise/background or extreme outliers. Without more robust metrics, it is challenging to adequately evaluate the models and results.

Finally, a major advantage of multiplexed imaging, specifically CyCIF, over other spatial techniques lies in its ability to capture morphological features in addition to single-cell intensities. Without showcasing the images generated by ME-VAE, it is difficult to determine if this model accurately represented the true features of these markers.

Minor comments

(1) Authors mentioned the data was split in 90%/5%/5% for training, validation and test sets. Why 90%, potential overfitting?

(2) While testing on WSI, a 0.26 reduction in Spearman correlation as authors called "modest performance reduction". Need more clarifications here.

Reviewer #2 (Remarks to the Author):

In this study, the authors translated a concept – masked language modeling, which is widely used in the context of natural language processing, into immuno-oncology. The modified version, masked image modeling (MIM), is capable of accurately impute unseen multiplexed imaging channels with minimal input data, thereby hold its promise in low-cost digital profiling of disease at single-cell level.

This study represents a novel application of AIML for biomarker discovery in the context of immuno-oncology and I personally encourage the paper to be published. With that said, I still have some concerns that need to be reconciled:

As I am not an expert in AI, I can only assess the paper from a computational biologist's point of view:

Major:

1. The authors denote the imaging data that they analyzed as 'CyCIF', however, this is not correct since the CRC cohort was collected using CODEX (CO-Detection by indEXing), which is different from CyCIF (cyclic immunofluorescence). One major difference is that CODEX uses DNA-barcoded antibodies to tag specific proteins whereas CyCIF uses iterative rounds of staining with fluorescently-labeled antibodies. The authors should reword to something like 'multiplexed tissue imaging'.
2. I commend the authors try to validate their method using images from different sources, however, I wonder if the model can be generalized across imaging modalities including IMC (imaging mass cytometry) and mIHC (multiplexed immunohistochemistry). The authors should at least include this in the discussion.
3. It is a bit unclear to me which panel yields the optimal results (perhaps 9 marker panel?). The authors should explicitly indicate this as well as names of markers in the panel.
4. For the most of the time, a single-cell dataset with phenotype annotated is the final readout from the image analysis and the input for downstream analysis, therefore I am wondering to what degree a cell phenotyping with simulated/generated channels can resemble real conditions? It would be very interesting if the authors could try such an experiment on at least one core.

Minor:

1. Add most recent refs that using various multiplexed imaging modalities for immune-oncology research:

mIHC – Pancreatic Cancer:

Mi, Haoyang, et al. "Quantitative spatial profiling of immune populations in pancreatic ductal adenocarcinoma reveals tumor microenvironment heterogeneity and prognostic biomarkers." *Cancer research* 82.23 (2022): 4359-4372.

Sequential IHC – Bladder cancer

Mi H, Bivalacqua T J, Kates M, et al. Predictive models of response to neoadjuvant chemotherapy in muscle-invasive bladder cancer using nuclear morphology and tissue architecture[J]. *Cell Reports Medicine*, 2021, 2(9).

IMC – lung adenocarcinoma

Sorin, Mark, et al. "Single-cell spatial landscapes of the lung tumour immune microenvironment." *Nature* 614.7948 (2023): 548-554.

MIBI – Breast Cancer

Keren L, Bosse M, Marquez D, et al. A structured tumor-immune microenvironment in triple negative breast cancer revealed by multiplexed ion beam imaging[J]. *Cell*, 2018, 174(6): 1373-1387. e19.

2. Unrecognized symbols (squares): lines 156 – 157.

Dear Reviewers,

On behalf of the authors, I wish to extend our sincere thanks for your time and effort focused on improving our work. All comments from each of the reviewers are presented below along with our response indicating how we incorporated feedback into our manuscript. Please find attached an updated manuscript with revised changes which have all been tracked.

Reviewers' comments:

Referee #1: multiplexed single-cell imaging, multi-channel imaging

Reviewer #1: *CyCIF and other multiplexed techniques offer the potential to measure numerous biomarkers, yet its panel capacity faces constraints due to technical obstacles. To address this limitation, the authors here introduced a computational method for panel reduction. This approach can effectively deduce information from markers employing just 9 key markers. It accomplishes this by learning co-expression and morphological patterns, resulting in enhanced speed, expanded panel content, and reduced costs. Though the idea and implementation here are both legit and solid, there are serious issues here need to be addressed.*

We thank the reviewer for the overall positive feedback and have addressed the reviewer's concerns in the following.

Comments:

Q1. *First, the evaluation of model performance was conducted solely at the single-cell level, lacking a population-scale assessment. Furthermore, since the primary purpose of CyCIF and other multiplex imaging approaches is to investigate spatial interactions and structures within tissues, modeling errors at the single-cell level could potentially magnify in these aspects. The authors should endeavor to address these caveats, such as comparing the overall distribution of markers between predicted and actual values.*

We thank the reviewer for raising this point. This is a valid concern.

In our prior research (Ternes et al. 2022), we showcased a population-scale assessment using cluster matching measured by normalized mutual information. In this current study, we present advancements in upstream performance that are anticipated to yield downstream benefits, specifically enhancing population-scale assessment and spatial interaction. This improvement stems from a reduction in modeling errors at the single-cell level within our proposed approach.

It is also noteworthy that a recent investigation by Wu et al. (2023) achieved promising outcomes, exhibiting an average Pearson correlation coefficient (PCC) of 0.534 for all predicted biomarkers. Their study emphasized the significance of imputed biomarkers in accurately classifying cell types, as evidenced by an F1 score of 0.727, and predicting patient survival outcomes. Our objective is to further elevate our model's performance, aiming for a correlation score of 0.9, surpassing existing standards. This aspiration is grounded in our results, which have already demonstrated performance that exceeds accepted benchmarks for biomarker prediction accuracy.

Ultimately, we believe that our contribution lies mostly within the model development aspect of this paper, as we propose a novel framework for streamlining the panel selection and marker imputation process. We agree that additional validation on downstream performance is necessary before adopting this method into traditional biological studies. However, we leave this to future work, partially because it is non-trivial to assemble a dataset that can serve as a sufficient benchmark. We have added this point to the discussion section.

Q2. *Secondly, the use of Spearman correlation here may not be entirely appropriate. Predicted intensities could differ significantly from the ground truth while still achieving a high Spearman*

correlation. Moreover, this correlation might be influenced by noise/background or extreme outliers. Without more robust metrics, it is challenging to adequately evaluate the models and results.

In response to the comment regarding the use of Spearman correlation, we acknowledge the concern raised and appreciate the opportunity to provide clarification. In our evaluation of mean marker intensity predictions, Spearman correlation was chosen to compare each marker's mean intensity between the ground truth and reconstructed images. We conducted this analysis individually for each stain and reported the average correlation across all stains in the set.

Within a single stain, downstream performance should not be affected if the absolute difference between the real and predicted mean intensities is large as long as the monotonic relationship between cells is preserved. Due to the inherent variability in staining intensities resulting from batch effects, staining intensity values should be treated as ordinal by nature. Therefore, Spearman correlation should be considered to be a sufficient metric for this task.

To address potential concerns about the sensitivity of Spearman correlation to differences in predicted intensities, we computed the correlation variance across stains. This approach allows us to assess the consistency of stain predictions, ensuring that the model performs well across all stains rather than excelling in some while performing poorly in others. By considering correlation variance, we aim to provide a more nuanced understanding of the model's predictive capabilities.

Furthermore, we acknowledge the importance of using robust metrics in our evaluation. In addition to Spearman correlation, we incorporated the Structural Similarity Index Measure (SSIM) to assess the quality of reconstructed CyCIF images. SSIM is a standard metric for tasks such as image translation, denoising, and restoration, as it evaluates both intensity and spatial information. This dual evaluation approach enhances the comprehensiveness of our assessment, addressing concerns about potential noise, background interference, or extreme outliers influencing the results.

In summary, we have already implemented a multi-faceted evaluation strategy, combining Spearman correlation with correlation variance and SSIM, to provide a more comprehensive and robust assessment of our models and results. It is worth highlighting that Spearman correlation and SSIM are commonly employed for assessing virtual staining in numerous literature sources including our previous works (Burlingame *et al.* 2021, Ternes *et al.* 2022).

Q3. Finally, a major advantage of multiplexed imaging, specifically CyCIF, over other spatial techniques lies in its ability to capture morphological features in addition to single-cell intensities. Without showcasing the images generated by ME-VAE, it is difficult to determine if this model accurately represented the true features of these markers.

We appreciate the reviewer's attention to the importance of capturing morphological features in addition to single-cell intensities in multiplexed imaging, specifically CyCIF. To address this concern, we incorporated the Structural Similarity Index Measure (SSIM) as a part of our evaluation metrics. SSIM is a widely recognized metric for assessing image quality, including the preservation of both intensity and spatial information. By utilizing SSIM, we aimed to ensure that our MAE model accurately represented the true features of the markers in the reconstructed images. Additionally, we have added a new figure (Figure 3) that showcases the reconstructed image channels.

Minor comments

(1) Authors mentioned the data was split in 90%/5%/5% for training, validation and test sets. Why 90%, potential overfitting?

We appreciate the reviewer's inquiry regarding the data split for training, validation, and test sets. The decision to allocate 90% for training was made to facilitate a comparison to our prior work (Ternes *et al.* 2022) on the same breast cancer (BC) TMA dataset, which was done with a

90%/10% train-test split. We make the additional split of the test set to produce results for the marker imputation step that is independent of the panel selection step. While we acknowledge the concern about potential overfitting, the purpose of the BC TMA dataset in this study is solely to benchmark the MAE performance against ME-VAE from Ternes *et al.* We conduct a more rigorous evaluation of the colorectal cancer (CRC) TMA dataset by using multiple train/test splits: the 90%/5%/5% split for comparison with the BC TMA results, and a 5-foldcross validation split (which equates to an 80%/10%/10% split), which was done at the TMA core level to demonstrate inter-patient generalization.

(2) While testing on WSI, a 0.26 reduction in Spearman correlation as authors called "modest performance reduction". Need more clarifications here.

We appreciate the reviewer's attention to the observed reduction in Spearman correlation during testing on Whole Slide Images (WSI). In our assessment of the performance, we acknowledge the noted 0.26 reduction in Spearman correlation, which we described as a "modest performance reduction."

To provide further context and clarification, we would like to highlight a relevant comparison with a recent study by Wu *et al.* (2023). In their investigation, they achieved promising outcomes with an average Pearson correlation coefficient (PCC) of 0.534 for all predicted biomarkers. This context underscores the challenges in achieving consistently high correlation values across different datasets and methodologies.

Furthermore, in the discussion, we have acknowledged that a single TMA is not representative of the full expression ranges that would be found in a WSI, as shown in Lin *et al.* 2023. We have emphasized this limitation and indicated that our future work will incorporate a larger dataset consisting of WSIs.

Reviewers' comments:

Referee #2: image informatics, tumor microenvironment, digital pathology

Reviewer #2: *In this study, the authors translated a concept – masked language modeling, which is widely used in the context of natural language processing, into immuno-oncology. The modified version, masked image modeling (MIM), is capable of accurately impute unseen multiplexed imaging channels with minimal input data, thereby hold its promise in low-cost digital profiling of disease at single-cell level.*

This study represents a novel application of AIML for biomarker discovery in the context of immuno-oncology and I personally encourage the paper to be published. With that said, I still have some concerns that need to be reconciled:

As I am not an expert in AI, I can only assess the paper from a computational biologist's point of view:

We thank the reviewer for the overall positive feedback. The concerns raised by the reviewer have been duly addressed in the subsequent sections.

Major:

Q1. *The authors denote the imaging data that they analyzed as 'CyCIF', however, this is not correct since the CRC cohort was collected using CODEX (CO-Detection by indEXing), which is different from CyCIF (cyclic immunofluorescence). One major difference is that CODEX uses DNA-barcoded antibodies to tag specific proteins whereas CyCIF uses iterative rounds of staining with fluorescently-labeled antibodies. The authors should reword to something like 'multiplexed tissue imaging'.*

In response to the reviewer's suggestion to reword the denotation to 'multiplexed tissue imaging,' we respectfully maintain that our original description of the imaging data as 'CyCIF' is accurate. We acknowledge the differences between CODEX and CyCIF, particularly the use of DNA-barcoded antibodies in CODEX and iterative rounds of staining with fluorescently labeled antibodies in CyCIF.

However, we would like to clarify that the imaging data utilized in our study is indeed from CyCIF, specifically the CRC TNP-TMA data from Lin *et al.* (2023). The confusion surrounding the use of CODEX does not apply in the context of our research, as we have consistently and correctly denoted the source of our imaging data.

Q2. *I commend the authors try to validate their method using images from different sources, however, I wonder if the model can be generalized across imaging modalities including IMC (imaging mass cytometry) and mIHC (multiplexed immunohistochemistry). The authors should at least include this in the discussion.*

Regarding your inquiry about the potential generalization of our model across various imaging modalities, including IMC (imaging mass cytometry) and mIHC (multiplexed immunohistochemistry), we highly value your suggestion. We agree that addressing the applicability of our model to diverse imaging techniques is a crucial aspect. In fact, in another ongoing study, we are currently evaluating the performance of our model on a different MTI platform, specifically the RareCyte Orion, to assess its generalizability.

It is noteworthy to mention that while our model's generalizability extends to other MTI platforms, we also acknowledge certain challenges associated with IMC and mIHC. IMC may face difficulties in utilizing morphological features effectively due to its resolution not being comparable to CyCIF. Additionally, mIHC's reliance on enzyme reactions introduces complexities in normalizing signal intensity, particularly in the absence of control tissue. These considerations have been thoughtfully incorporated into the discussion section of our revised manuscript.

Q3. *It is a bit unclear to me which panel yields the optimal results (perhaps 9 marker panel?). The authors should explicitly indicate this as well as names of markers in the panel.*

We apologize if there was any ambiguity regarding the optimal marker panel and the specific markers used in our study. In response to your concern, we would like to direct your attention to Figure 2A (breast cancer) and Supplementary Figure 1 (Colorectal cancer), where we have provided details on marker orders. We believe that the marker panel yielding optimal results, including the names of markers, is presented in these figures.

Q4. *For the most of the time, a single-cell dataset with phenotype annotated is the final readout from the image analysis and the input for downstream analysis, therefore I am wondering to what degree a cell phenotyping with simulated/generated channels can resemble real conditions? It would be very interesting if the authors could try such an experiment on at least one core.*

We thank the reviewer for raising this point. This is a valid concern.

In our prior research (Ternes *et al.* 2022), we showcased a population-scale assessment using cluster matching measured by normalized mutual information. In this current study, we present advancements in upstream performance that are anticipated to yield downstream benefits, specifically enhancing population-scale assessment and spatial interaction. This improvement stems from a reduction in modeling errors at the single-cell level within our proposed approach (also see our response to Reviewer 1 Q1).

While downstream performance is certainly an important factor in model evaluation, cell phenotyping performance can be difficult to assess. For example, the cell type labels utilized in Wu *et al.* were derived by manually assigning labels to clusters of mean intensity values in principal component space. Because these labels are ultimately arbitrary and predicated by the mean intensity values anyway, we believe that measuring mean intensity correlation is the most reliable way to quantify model performance in this context.

Minor:

1. Add most recent refs that using various multiplexed imaging modalities for immune-oncology research:

- *mIHC – Pancreatic Cancer: Mi, Haoyang, et al. "Quantitative spatial profiling of immune populations in pancreatic ductal adenocarcinoma reveals tumor microenvironment heterogeneity and prognostic biomarkers." Cancer research 82.23 (2022): 4359-4372.*
- *Sequential IHC – Bladder cancer: Mi H, Bivalacqua T J, Kates M, et al. Predictive models of response to neoadjuvant chemotherapy in muscle-invasive bladder cancer using nuclear morphology and tissue architecture[J]. Cell Reports Medicine, 2021, 2(9).*
- *IMC – lung adenocarcinoma Sorin, Mark, et al. "Single-cell spatial landscapes of the lung tumour immune microenvironment." Nature 614.7948 (2023): 548-554.*
- *MIBI – Breast Cancer Keren L, Bosse M, Marquez D, et al. A structured tumor-immune microenvironment in triple negative breast cancer revealed by multiplexed ion beam imaging[J]. Cell, 2018, 174(6): 1373-1387. e19.*

In response to your recommendation, we have thoroughly reviewed the literature and identified pertinent recent studies. We have promptly incorporated these relevant references into the revised manuscript to provide a more comprehensive and up-to-date overview of the current state of research in immune-oncology utilizing multiplexed imaging modalities.

2. *Unrecognized symbols (squares): lines 156 – 157.*

We apologize for the issue encountered during the conversion of our submission from a Word file to a PDF. To address this, we will submit our revised version directly in PDF format, ensuring a smooth and error-free submission.

Reviewers' comments:

Reviewer #1 (Remarks to the Author):

In the updated manuscript, the authors have made substantial improvements in response to my previous concerns. Notably, the inclusion of SSIM analysis has bolstered the credibility of their findings. Nevertheless, I have identified a potential issue with the newly introduced Figure 3. Several of the images or markers presented in this figure do not appear to align with established prior knowledge. For instance, DAPI, which is expected to serve as a nuclear marker, and CD45, typically associated with membrane staining, appear remarkably similar in these images. This raises questions about the integrity of the training data used. It would be advisable for the authors to revisit and revise these panels, selecting the most representative images to ensure accuracy and consistency.

Reviewer #2 (Remarks to the Author):

The authors have addressed all concerns. The manuscript looks good to me.

Dear Reviewers,

Thank you for your thorough review of our manuscript and for acknowledging the substantial improvements we have made in response to previous review comments. We appreciate your valuable feedback and the opportunity to address your comments.

Reviewer #1 (Remarks to the Author):

In the updated manuscript, the authors have made substantial improvements in response to my previous concerns. Notably, the inclusion of SSIM analysis has bolstered the credibility of their findings. Nevertheless, I have identified a potential issue with the newly introduced Figure 3. Several of the images or markers presented in this figure do not appear to align with established prior knowledge. For instance, DAPI, which is expected to serve as a nuclear marker, and CD45, typically associated with membrane staining, appear remarkably similar in these images. This raises questions about the integrity of the training data used. It would be advisable for the authors to revisit and revise these panels, selecting the most representative images to ensure accuracy and consistency.

Response: We have carefully considered the concerns you raised regarding Figure 3 in the updated manuscript. Specifically, we understand your observations regarding the discrepancies between the expected staining pattern of certain markers, such as CD45 and their appearance in the images presented.

This discrepancy primarily arises from the selection of a single exemplar image that inaccurately represents our dataset. Firstly, the image was cropped, and the background was zeroed out without adjusting the contrast, leading to a misleading appearance. Here, we provide images from our dataset (see Figure R1), specifically showcasing DAPI and CD45, which accurately depict the expected staining patterns of these markers. As all the data utilized in this study is publicly accessible, we have included a link for the reviewer to explore the dataset. In our revision, we have updated Figure 3 with improved exemplar images depicting CD45 expression.

Figure R1. Example crops of breast cancer TMA cores (DAPI in blue and CD45 in green). CD45 staining pattern clearly shows membrane staining pattern. These images are available at the HTAN website:

https://d3p249wtgzkn5u.cloudfront.net/synid/syn26341508/minerva/index.html#s=0#w=0#g=6#m=-1#a=-100_-100#v=1.3396_0.4188_0.5854#o=-100_-100_1_1#p=Q#r=100

Additionally, we provide example cells below to demonstrate the compartmentalization of DAPI versus CD45. There is a distinct region on the periphery of each cell that is positive only for CD45, as expected, with DAPI localized to the center of the cell.

We have revised Figure 3 (see revised manuscript):

Furthermore, this paper presents computational methodologies. As we continue to refine the data by removing artifacts and addressing staining issues, we anticipate further enhancements in our predictive capabilities.